# Self-healable printed magnetic field sensors using alternating magnetic fields

Rui Xu[1] ✉, Gilbert Santiago Cañón Bermúdez [1], Oleksandr V. Pylypovskyi [1,2], Oleksii M. Volkov [1], Eduardo Sergio Oliveros Mata [1], Yevhen Zabila[1], Rico Illing[1], Pavlo Makushko[1], Pavel Milkin[3], Leonid Ionov[3], Jürgen Fassbender [1] & Denys Makarov [1] ✉

We employ alternating magnetic fields (AMF) to drive magnetic fillers actively and guide the formation and self-healing of percolation networks. Relying on AMF, we fabricate printable magnetoresistive sensors revealing an enhancement in sensitivity and figure of merit of more than one and two orders of magnitude relative to previous reports. These sensors display low noise, high resolution, and are readily processable using various printing techniques that can be applied to different substrates. The AMF-mediated self-healing has six characteristics: 100% performance recovery; repeatable healing over multiple cycles; room-temperature operation; healing in seconds; no need for manual reassembly; humidity insensitivity. It is found that the above advantages arise from the AMF-induced attraction of magnetic microparticles and the determinative oscillation that work synergistically to improve the quantity and quality of filler contacts. By virtue of these advantages, the AMF-mediated sensors are used in safety application, medical therapy, and human-machine interfaces for augmented reality.

Wearable electronics are crucial for ongoing technological trends like the internet of things and augmented reality, as they communicate with other gadgets and react to their surroundings by integrating numerous sensors fabricated using printing methods and flexible electronic modules[1–4]. For achieving high performance and extending the lifespan of printable wearable sensors, it is imperative to form and reform (i.e., self-heal) percolation networks of functional fillers in the composite[5,6]. Besides establishing a percolation path, low cost, scalability, and printability should also be maintained in industrial production[7,8]. Six additional demands also need to be met for practical self-healing: (1) 100% performance recovery; (2) repeatable healing over multiple cycles; (3) fast healing to shorten the time of functional loss; (4) room-temperature operation to avoid harming surrounding materials; (5) independence of ambient condition, e.g., humidity; (6) no need for manual reassembly to meet with special cases where manual touch is impossible or time-consuming (e.g., for implantable

electronics)[9–11]. Simultaneously satisfying all these requirements poses a major challenge.

So far, tactile printable sensors that are robust and self-healing have been demonstrated[12–16]. However, prospective interactivity will rely more on touchless interfaces[17–19], e.g., to avoid the spread of viruses/bacteria in the post-COVID-19 world and to offer three-dimensional immersive experience for customers. In contrast to touchless interplay enabled by optical and electrical (capacitive) sensors[20,21], magnetic field sensors are immune to environmental disturbances (e.g., opaque obstacles and humid conditions), and thus represent a promising solution to this request[22]. Despite significant progress, realizing high magnetoresistive response (i.e., large change of the electrical resistance in a magnetic field) with low noise and high sensing resolution is still challenging in printable magnetic field sensors, especially at human-safe magnetic fields, and even more so, if combined with self-healing[23]. The main difficulty has been that

[1]Helmholtz-Zentrum Dresden-Rossendorf e.V., Institute of Ion Beam Physics and Materials Research, Bautzner Landstrasse 400, 01328 Dresden, Germany. [2]Kyiv Academic University, Kyiv 03142, Ukraine. [3]Bavarian Polymer Institute, University of Bayreuth, Ludwig Thoma Str 36a, 95447 Bayreuth, Germany. ✉e-mail: r.xu@hzdr.de; d.makarov@hzdr.de

methods established towards percolation network (re)formation of tactile sensors are not practically applicable to printable magnetoresistive sensors[24–27], e.g., increasing filler concentration, introducing conductive additives, and engineering fillers with nanoscaled features will deteriorate mechanical stability, shunt magnetic performance, and raise synthesis complexity (cost) of composite. Therefore, an approach, which directly improves the quality and quantity of physical contacts between magnetoresistive fillers and meanwhile adapts to practical production and self-healing, is urgently needed.

In this work, we successfully addressed this issue by employing alternating magnetic field (AMF) to manipulate magnetoresistive fillers (here, $Ni_{81}Fe_{19}$ microparticles). Printable sensors made of the AMF-mediated composites achieved superior magnetoresistive performance, i.e., sensitivity of $35.7\,T^{-1}$ at 0.086 mT, figure of merit (FoM) of $4.1 \times 10^5\,T^{-2}$, low noise of 19 $\mu\Omega/\sqrt{Hz}$, and high resolution of 36 nT. The sensitivity and FoM were improved by at least one and two orders of magnitude as compared to the reported ones, respectively, and the magnetic field was far below the 40 mT limit of long-lasting exposure prescribed by the World Health Organization (WHO)[28]. Importantly, the high performance was independent of the applied printing techniques and substrates thanks to the determinant role of AMF in percolation network formation, thus suitable for practical production. With the help of AMF, the healing process can be finished in seconds at room temperature regardless of humidity (even in water) and did not need manual reassembly. The magnetoresistive performance suffered no degradation after multiple damage/self-healing cycles. Both

simulation and experiments prove that the above superiorities arise from the controllable attraction and oscillation of magnetic fillers in response to AMF that account for compact arrangement and close contact, respectively, and work synergistically for percolation network (re)formation. These AMF-mediated advantages pave the way for the printable magnetoresistive sensors, which can serve as assistive devices for bio-monitoring, smart textiles for safety applications, medical therapy, and human–machine interfaces for augmented reality.

## Results

### AMF-mediated fabrication and self-healing of printable magnetoresistive sensor

The fabrication process of AMF-mediated printable magnetoresistive sensors is schematically illustrated in Fig. 1a, which can be generally divided into composite synthesis, printing, and curing. For the mixture binder, biocompatible polydimethylsiloxane (PDMS) served as an architectural scaffold to enable stable structure, and viscoelastic supramolecular polyborosiloxane (PBS) provided self-healing capability because of the dynamic boron/oxygen dative bonds between neighboring chains and entanglement of polymeric chains[29–31]. To confirm practical viability of the AMF-mediated method, we used commercial $Ni_{81}Fe_{19}$ microparticles as fillers, rather than the widely adopted nanostructures that are easier for percolation but difficult (expensive) in synthesis. Microparticles feature spherical shape, smooth surface, and irregular size (Supplementary Fig. 1) and the majority are about 3–5 μm in diameter (Supplementary Fig. 2).

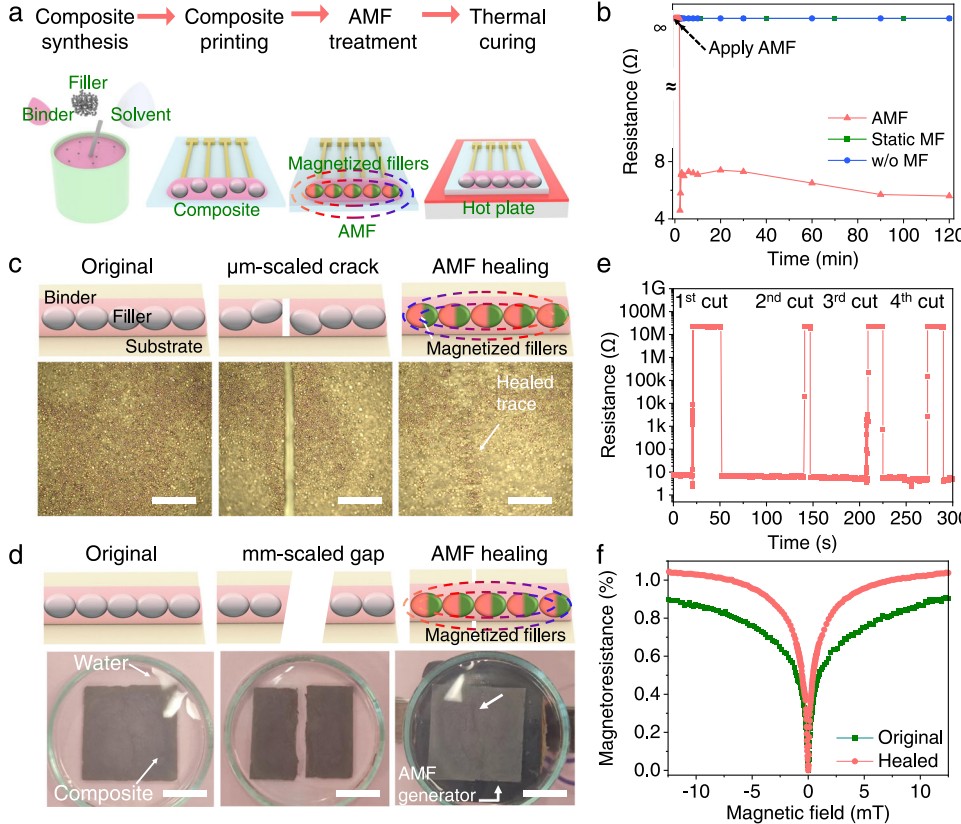

**Fig. 1 | Fabrication and self-healing of magnetoresistive sensors aided by AMF.**
**a** Schematic illustration of fabrication process for magnetoresistive sensors.
**b** Electrical resistance of magnetoresistive composite during curing.
**c, d** Photographs and schematics of the magnetoresistive composite (left to right: original, cutting, healing by 20-s long AMF exposure). Magnetized $Ni_{81}Fe_{19}$ microparticles lead to an active healing of percolation networks without the need for manual assembly. In (**c**), the magnetoresistive composite was damaged with micrometer-scaled cracks. In (**d**), the composite was disconnected with millimeter-

scaled gaps and healed in water. Please see Supplementary Movies 1, 3 for details. Scale bars: 400 μm in (**c**) and 10 mm in (**d**). **e** Resistance variation during four cutting/self-healing cycles. The actual time for self-healing was shorter than that displayed, considering a major fraction of no-conductivity time required to generate damage and place into the AMF setup, as observed in Supplementary Movie 2. **f** Magnetoresistance before and after cutting/healing. The magnetoresistance was characterized by measuring the variation of electrical resistance in a tunable magnetic field, normalized to the original resistance at 0 mT, that is, $(R-R_0)/R_0$.

The cured composites exhibit higher storage moduli than loss moduli over the whole range of frequency (Supplementary Fig. 3), revealing an elastic behavior and mechanical stability (i.e., no flowing) over long time scales which is in agreement with the previous reports[29,32,33]. When applying AMF (of 50 Hz frequency and <130 mT amplitude, see Supplementary Fig. 4) to composite during curing, the electrical resistance decreased sharply to several ohms in seconds (Fig. 1b). The phenomenon was observable in composites of different $Ni_{81}Fe_{19}$ concentrations, achieving an averaged electrical conductivity of 5.4 S/cm (Supplementary Fig. 5). For comparison, the bare composite without magnetic field treatment and the composite treated with the static magnetic field have no electrical conductance (Fig. 1b), although the applied 500 mT was much higher than the above AMF. To assess the AMF-mediated percolation network in magnetoresistive performance, we took a $Ni_{81}Fe_{19}$ based sensor prepared with thin film technologies as benchmark (e.g., displaying 1.89% magnetoresistance)[34]. The printed AMF-mediated sensor (with 0.91% magnetoresistance at 12.5 mT) reached 48.1% of the performance of the thin film counterpart; while the nanoflake-based sensor without AMF treatment (with 0.1% magnetoresistance at 17 mT) only had 5.3%[32], even though nanoflakes with larger surface ratios and longer electrical continuity are more desirable for electrical percolation[35].

To investigate the AMF-mediated self-healing capability, both micrometer-scaled cracks and millimeter-scaled gaps were introduced to disconnect the electrical path of magnetoresistive composite. For a 100-µm-wide crack (Fig. 1c), $Ni_{81}Fe_{19}$ microparticles oscillated in AMF and viscoelastic polymeric binders flew along (Supplementary Movie 1). The AMF-mediated reestablishment of electrical percolation was directly confirmed by a circuit composed of a magnetoresistive sensor and a commercial light-emitting diode (LED) (Supplementary Movie 2, Supplementary Fig. 6). Besides reforming conductive pathways like that in conventional methods (e.g., by taking advantage of the mobility of polymers which drives the movement of fillers)[9,11], the AMF-mediated self-healing could 100% recover the original electrical property in few seconds due to the controllable magnetic force (e.g., strength and direction of the applied magnetic field, its frequency, and the actuation time) applied to $Ni_{81}Fe_{19}$ microparticles. In particular, the AMF-generated force is independent of the surrounding conditions (e.g., temperature and humidity), thus broadening the applicability of the AMF-induced self-healing. To confirm the repeatability of the AMF-mediated self-healing, we carried out four cutting/healing cycles and found that the corresponding resistance had no increase and even reduced from 7.5 to 5.4 ohms (Fig. 1e), ascribed to spatial rearrangement of $Ni_{81}Fe_{19}$ microparticles and tightened physical contact. The enhancement in electrical percolation gave rise to magnetoresistance improvement, e.g., from 0.9% to 1.04% (Fig. 1f). While without the AMF treatment, self-healing of electrical percolation was not observed at room temperature, probably because the mobility of viscoelastic polymer matrix could not provide strong enough force to move $Ni_{81}Fe_{19}$ microparticles in consideration of their dense and heavy nature. When being healed with a thermal heating method at 120 °C, the electrical percolation path was reformed, however, the corresponding magnetoresistance was reduced from 0.92% to 0.64% due to the loosened contact during thermal expansion[11,36], as proved by the resistance increase from 8.1 to 20.8 Ω (Supplementary Fig. 7). It is important to notice that, only a minor faction of samples reformed the electrical path even heating at such high temperatures. In contrast, the AMF-mediated self-healing can be easily fulfilled at room temperature and does not need thermal heating to strengthen polymer (and thus filler) mobility[37]. The healed sensor had reliable operational stability, showing little resistance variation and magnetoresistance degradation in time (Supplementary Fig. 8). The AMF-mediated method also adapted to the reparation of splitting with millimeter-sized gaps even in humid environments (Fig. 1d). In Supplementary Movie 3, two magnetoresistive composites, soaked in water, were able to reconnect

automatically within several milliseconds only with the assistance of the attracting force generated in the magnetized $Ni_{81}Fe_{19}$ microparticles, beneficially omitting manual reassembly of disconnected composites[9,11]. In stark contrast to the devices that incorporate permanent magnets as fillers for automatic reconnection in self-healing, the AMF-mediated self-healing sensor has no magnetic remanence and thus will not pose risk to human health and/or interfere with the functionality of nearby electronics[38,39]. In particular, as compared with the manual reconnection at the macroscale (during which microscopic gaps might be retained at the damaged interface), the strong attracting force triggered by the magnetized microparticles could result in an intimate adhesion at the microscopic domain, which is beneficial for 100% healing of the electrical performance. The maximum gap width with which two fragments can be attracted and healed depends on the applied AMF and thus is easily controllable.

## Magnetoresistive characterization of AMF-mediated printable sensors

To optimize magnetoresistive performance, we investigated the $Ni_{81}Fe_{19}$ microparticle concentration dependence and found that saturated magnetoresistance at 12.5 mT increased from 0.75% to 1.01% when raising the concentration from 0.15 g/ml to 1 g/ml (Fig. 2a), i.e., the volume fractions of the microparticles in composite vary from 6.7% to 36.5%. This change in the sensor performance arises from electrical percolation improvement, as experimentally proved by electrical conductivity in Supplementary Fig. 5. The concentration of $Ni_{81}Fe_{19}$ microparticles also affects the self-healing rate and the magnetoresistance performance (Supplementary Fig. 9). For instance, if the volume fraction of microparticles in the cured composite is lower than the ideal percolation threshold of about 16%[12,40], the healing of the damaged sensors was not reproducible. We anticipate that the reason for this poor self-healing performance is related to the presence of microscopic gaps at the damaged interface under weak attracting force and insufficient reconnection of percolation networks. With the addition of more microparticles to the composite (e.g., to 22.3% and 36.5% volume fractions), the damaged sensors can be healed easily in AMF with little magnetoresistance reduction. In the following experiments, magnetoresistive sensors were fabricated using the 1 g/ml concentration of $Ni_{81}Fe_{19}$ microparticles in the composite. A highest magnetoresistive sensitivity of 35.7 $T^{-1}$ was obtained at 0.086 mT (Fig. 2b). This operational magnetic field is lower than the WHO limit of 40 mT for continuous exposure[28], thus suitable for wearable use. The AMF-mediated sensors also possess small electrical noise at low magnetic fields, for example, the noise power density at a current of 1.24 mA decreased to 19 µΩ /√Hz at 1.8 Hz (Fig. 2c). In combination with the high sensitivity, a maximum resolution of about 36 nT can be estimated (Supplementary Fig. 10), allowing to differentiate sub-µT-scaled magnetic fields for precise operation.

Both the high sensitivity and low operational magnetic field induced by AMF outperform all the previous reports, as summarized in Fig. 2d. Compared to the best values realized in different types of printable magnetoresistive sensors, e.g., based on effects of giant magnetoresistance (using $[Ni_{81}Fe_{19}/Cu]_{30}$-nanoflakes fillers)[41], anisotropic magnetoresistance (using $Ni_{81}Fe_{19}$-nanoflakes fillers)[34], and tunneling magnetoresistance (using Fe-nanoparticles fillers)[42], the AMF-induced sensitivity was enhanced about 12, 18.8, and 1552 times, respectively. A figure of merit was defined for an overall evaluation of sensitivity and operational magnetic field for various sensors. The AMF-mediated printed sensor achieved unmatched 4.2 × $10^5$ $T^{-2}$, which had at least two orders of magnitude boost over previous reports. The high performance was observed for the sensors prepared with different printing techniques, e.g., pipetting, screen printing, and spin coating as well as target substrates including commercial flat flexible cables (FFC), Si wafers, glass slides, plastic, paper, and ceramic,

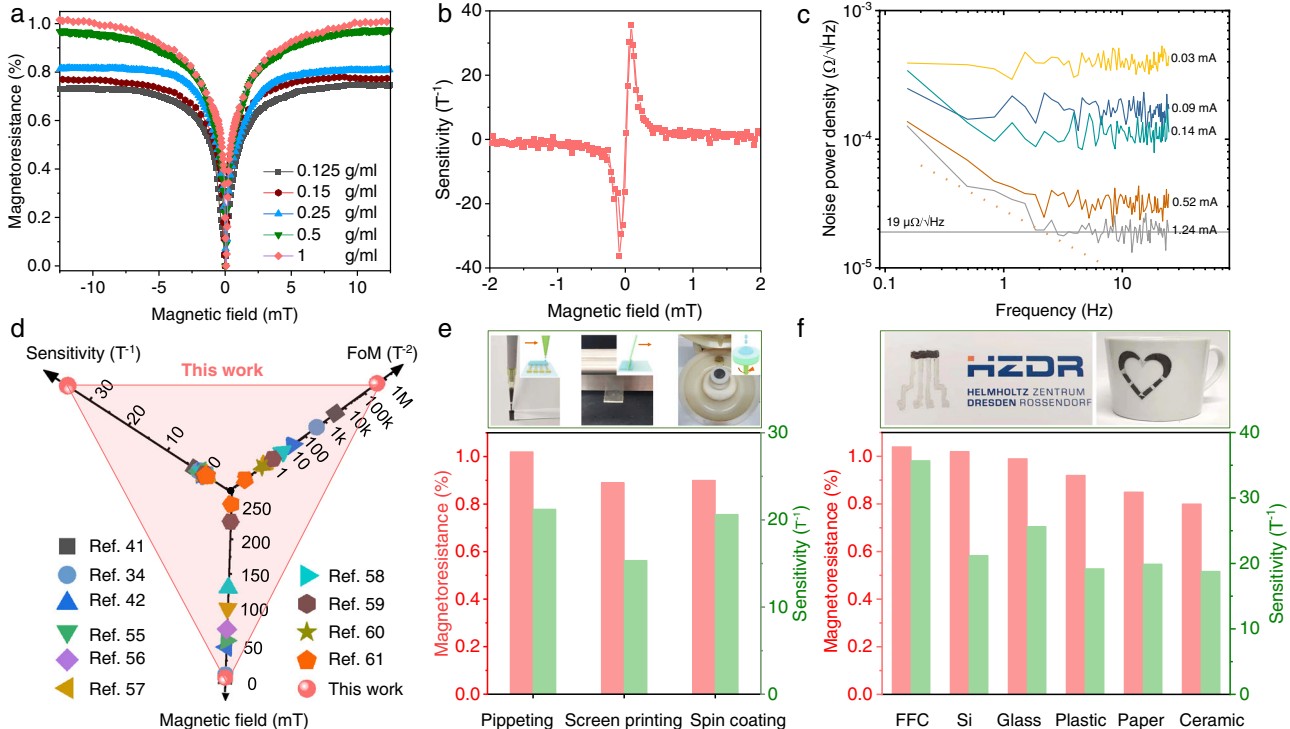

**Fig. 2 | Characterization of AMF-mediated printable magnetoresistive sensors.**
**a** Magnetoresistance of sensors made by composites of different $Ni_{81}Fe_{19}$ micro-
particle concentrations. **b** Magnetoresistive sensitivity of sensor of 1 g/ml con-
centration. Sensitivity was calibrated by the first derivative of electrical resistance
with respect to magnetic field of magnetoresistive element divided by resistance.
**c** Noise signal of sensor. **d** Comparison of sensitivity, magnetic field where the
highest sensitivity is obtained, and figure of merit (FoM) for printable magnetor-
esistive sensors in previous reports[34,41,42,55–61] and this work. Figure of merit is
defined as the maximum sensitivity divided by the magnetic field where maximum

sensitivity is obtained. The reported printable sensors were made of 0-dimensional
nanoparticles[42,55–57], 1-dimensional nanowires[60], and 2-dimensional
nanoflakes[34,41,55,59,61], and based on effects of anisotropic magnetoresistance[34], giant
magnetoresistance[41,55,58–61], and tunneling magnetoresistance[42,56,57]. **e** Photographs
and schematic illustrations of different fabrication techniques for printable sen-
sors, and the corresponding magnetoresistance and sensitivity.
**f** Magnetoresistance and sensitivity of sensors fabricated on different substrates.
Photographs exhibit sensors printed on paper and ceramic cup.

retaining the magnetoresistive response of >0.89% and sensitivity of
>15.4 T$^{-1}$, as summarized in Fig. 2e, f and Supplementary Figs. 11, 12.

## Mechanism of (re)forming electrical percolation between fillers by AMF

Magnetic and electric simulations were carried out to unveil the
underlying mechanism of the AMF-mediated percolation formation.
Firstly, we investigated the static interaction of two filler micro-
particles (e.g., of soft magnetic $Ni_{81}Fe_{19}$) in response to magnetic fields.
Once exposed to magnetic fields, two separated microparticles (i.e.,
corresponding to the printed or damaged composite) are magnetized
immediately, generating enhanced magnetic fields around micro-
particles (especially in the gap volume) due to their high magnetic
permeability (Fig. 3a$_1$). Consequently, a strong attracting force is trig-
gered between microparticles (Fig. 3a$_2$), driving them to move towards
one another. However, the presence of the polymeric binder tends to
prevent physical contacts. Electrical simulation reveals that an insu-
lating binder layer as thin as 1 nm is capable of blocking the electrical
path between microparticles (Supplementary Fig. 13), which is in
accordance with the aforementioned reference samples and the pre-
vious reports that carrier transports barely occur between smooth
microparticles[12]. Besides the strong attracting force, a tangential force
is also exerted, especially after two microparticles get close (Fig. 3a$_3$,
Supplementary Fig. 14), which drives magnetic microparticles to align
with the external magnetic field. As the field value is alternating, this
force is altered accordingly, which is believed to cause the oscillation
of microparticles. The AMF-induced dynamic interaction is confirmed
theoretically by the time-varying behavior of a microparticles system,
as shown in Supplementary Movie 4. The motion of microparticles is

governed by the Newton equation

$$m\ddot{\mathbf{r}}_i = M_S \sum_{j \neq i} \int \nabla\left(\boldsymbol{\mu}_i \cdot \mathbf{B}_j(\mathbf{r}_i)\right)dV - 6\pi R\eta\dot{\mathbf{r}}_i, \; i = \overline{1,N} \qquad (1)$$

with overdot being the derivative with respect to time $t$, $\mathbf{r}_i$ being the
radius-vector of $i$th particle, $m$ being microparticle mass, the terms
on the right side being the magnetic force and Stoke's force,
respectively (see "Methods" for details). In AMF, the strongest
attracting forces for two parallel magnetic moments act towards
their poles (Fig. 3b$_1$). This distribution of the force lines favors the
formation of elongated chains with a weak inter-chain interaction
(Fig. 3b$_2$). With change of the field direction, each separate chain
reverses its total magnetic moment instantly (Supplementary
Fig. 15), resulting in the oscillation of microparticles relative to
one another (Fig. 3b$_3$). Actually, the microparticle chains can be
formed quickly (on the scale of µs) once applying AMF, and their
oscillation takes the most of the rest of time (Supplementary
Movie 4). Such oscillation could mill off a thin binder between
microparticles to establish electrical and physical contact. The
AMF-treated composite with elongated chains and close contacts is
desirable for electrical percolation. The theoretically predicted
attraction and oscillation of microparticles under AMF, as well as
the consequent formation of a percolation network, were experi-
mentally confirmed as shown in Supplementary Movie 5 and Fig. 3c.
Excluding the influence of the temperature increase over resistance
variation (Supplementary Figs. 16, 17)[43], we conclude that the AMF-
induced attraction and especially collision of the microparticles
play a determinative role for electrical percolation.

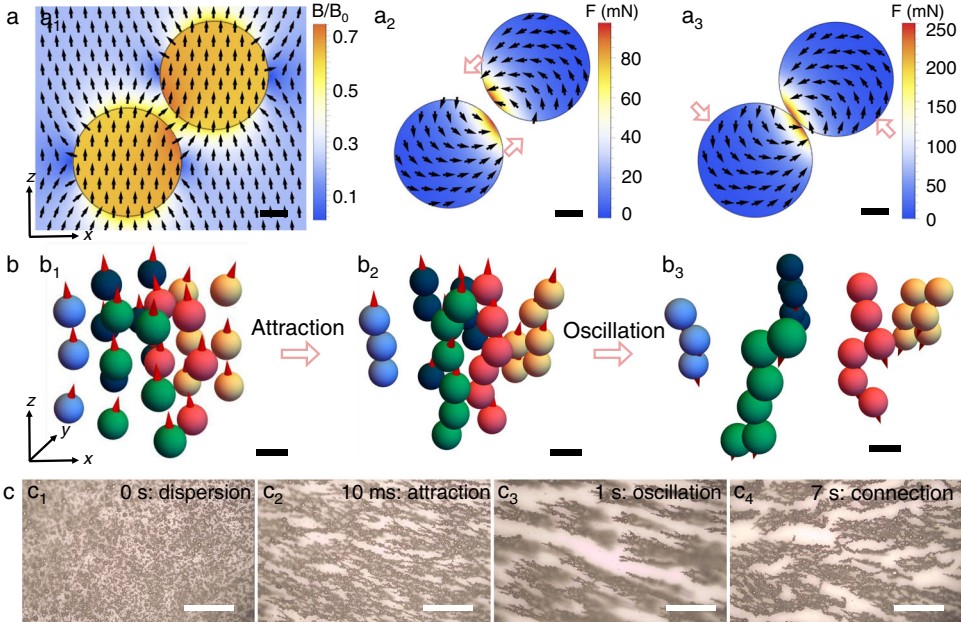

**Fig. 3 | Magnetic simulations of interaction of magnetic microparticles under AMF. a** Static simulations revealing interaction between two $Ni_{81}Fe_{19}$ microparticles exposed to a homogeneous external magnetic field. **a$_1$** Magnetic stray field distribution and **a$_2$** force distribution as two microparticles are apart, corresponding to a very early stage of applying AMF. $B_0$ indicates the saturation field of $Ni_{81}Fe_{19}$ microparticles. **a$_3$** Force distribution as two microparticles are in contact. Pink arrows qualitatively portray the main direction of the net force experienced by microparticles. Scale bars in a): 1 μm. **b** Dynamic simulations revealing interaction between 27 $Ni_{81}Fe_{19}$ microparticles exposed to AMF. Screenshots of microparticles interaction in Movie 1 after applying AMF for **b$_1$** 0 μs, **b$_2$** 30 μs, and **b$_3$** 500 μs. Particles' colors aid the eyes for tracking the time evolution of the particle's positions. Scale bars in **b**): 4 μm. Red arrows show the direction of magnetic moments. Coordinate reference frame is shown with black arrows in panel **d**. Magnetic fields are set to be along the $z$ direction. **c** Screenshots showing the real-time interaction of $Ni_{81}Fe_{19}$ microparticles under AMF in experiment: **c$_1$** dispersion; **c$_2$** attraction; **c$_3$** oscillation; **c$_4$** connection. The values on the top-right corner are the time since applying AMF. Scale bars: 400 μm.

## Applications of AMF-mediated printable magnetoresistive sensors

In light of the merits introduced above (e.g., self-healable, high sensitivity, safe operational magnetic field, low noise, high resolution, and practical fabrication), the AMF-mediated printable magnetoresistive sensors hold promise to provide personalized assistive devices. Here, we exhibit three envisaged examples.

### Wearable application for safety monitoring

The printed sensors mounted onto garments are able to detect small magnetic field variation, which can assist in safety applications. For example, the ubiquity of smart electronics (e.g., phone, wristband) that generate magnetic interference poses potentially serious risks to people implanted with pacemakers or cardioverter defibrillators due to possible malfunction (Supplementary Fig. 18)[44,45]. We divided the magnetoresistance range of the printed sensor into four bands, separated by three thresholds that triggered on green, yellow, and red LEDs, respectively. As a permanent magnet approached the sensor, magnetic fields surrounding the sensor intensified (Fig. 4a) and generated large resistance variation to cross the three magnetoresistance thresholds (Fig. 4b), turning on the indicating LEDs in sequence and indicating the invisible risk (Supplementary Movie 6). Depending on the magnetic stimulus (e.g., magnetic field intensity and its spatial distribution), the sensing zones are altered for different magnets. For example, the boundaries between Zone 1, Zone 2, Zone 3 and Zone 4 are 63, 50, and 27 cm (Supplementary Fig. 19b, c) as well as 46, 31, and 15 (Supplementary Fig. 19d, e) away from the corresponding magnets.

Considering that magnetic field might come from any direction, we carried out both in-plane and out-of-plane measurement and observed an isotropic magnetoresistance response (Supplementary Figs. 20, 21) in contrast to the anisotropy of the film-based counterpart (Supplementary Fig. 22), mainly ascribed to the random spatial

orientation of $Ni_{81}Fe_{19}$ microparticles in the composite[41]. Such omnidirectional performance and the AMF-induced robust filler contact work in concert to achieve a bending independence of magnetoresistance, for example, saturated magnetoresistance of over 0.89% was preserved and only 0.068% decrease was detected when deforming a planar sensor to the states with curvature radii of 5 to 1 mm (Fig. 4c). Besides, the AMF-treated sensor has reliable operational stability (Supplementary Fig. 23). After hundreds of bending/unbending cycles, the high magnetic sensing capability was maintained. Notably, the magnetoresistance suppression induced by the cyclic deformation can be easily healed by the AMF treatment. This isotropic magnetoresistance and operational durability of the sensor response allow for an easy operation in practical use that do not require pre-alignment of magnetic fields and pre-flattening of the sensors.

Importantly, the sensors feature waterproof performance because of the complete isolation of $Ni_{81}Fe_{19}$ microparticles from water by the polymeric binder (Supplementary Fig. 24) and possess identical magnetoresistance response in air and water (inset of Fig. 4d, Supplementary Fig. 25a, b). After immersing in water for four weeks, no degradation was observed in both electrical resistance and magnetoresistance (Fig. 4d). Such water resistance allows wearable magnetoresistive sensors to perform in broad circumstances, e.g., swimming pool and rainy days and leads to the viability of sensor-mounted garments that are functional and repairable in water (Fig. 4e). After damage, the magnetoresistive response can be healed even in water with the aid of AMF (Supplementary Fig. 25c, d), ascribed to the fast automatic reconnection, the intimate adhesion at the microscopic domain, and the enhanced mobility of the polymer chains.

### On-skin application for medical therapy and rehabilitation

The printed sensors can conformally adapt to human skin with morphologically different surfaces and suffer no delamination from

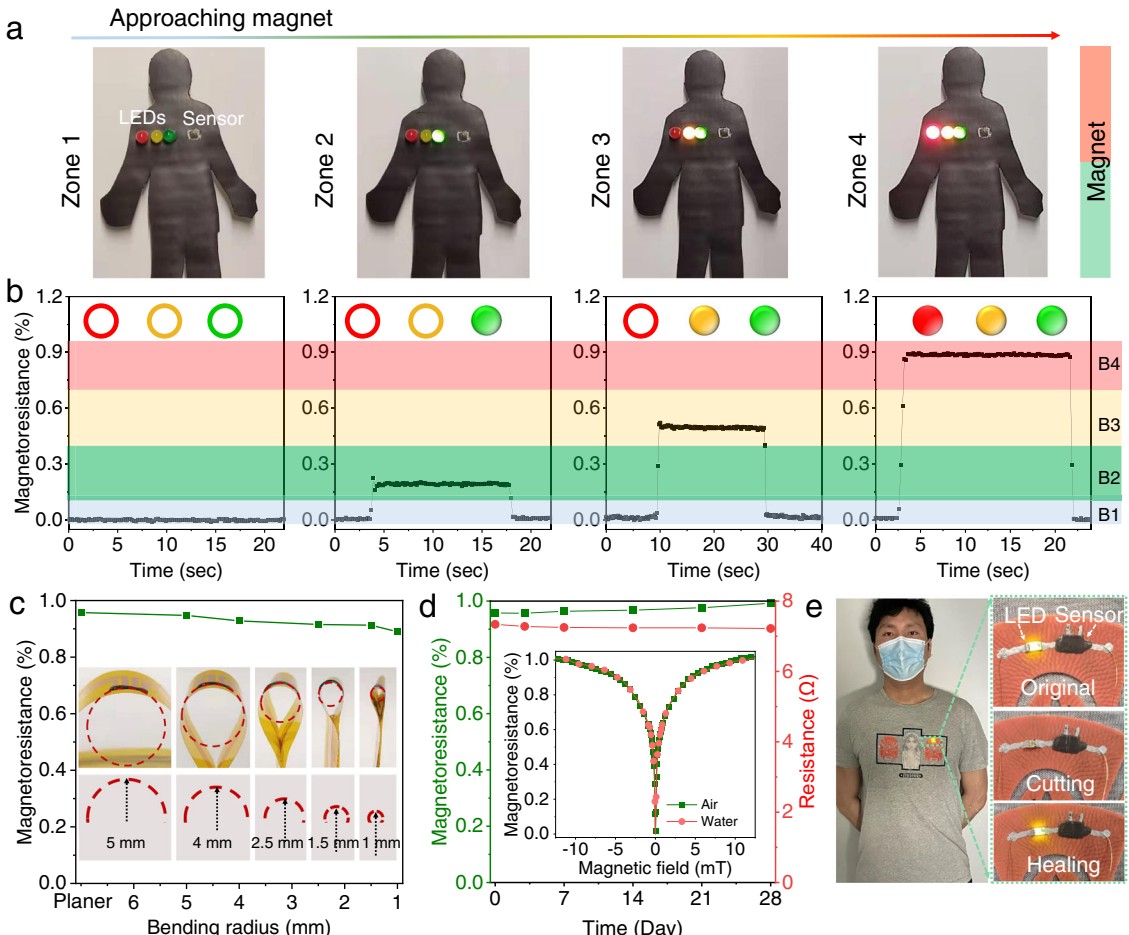

**Fig. 4 | Wearable printable magnetoresistive sensor mounted onto garments for safety application. a, b** Indicator of magnetic fields for patients implanted with magnetic field sensitive devices (e.g., cardioverter defibrillator and pacemakers). **a** Photograph of humanoid model decorated with green, yellow, and red LEDs controlled by magnetoresistance signal of sensor. As approaching a permanent magnet, the sensor entered into Zone 1–4 and triggered LEDs on one-by-one due to the intensified magnetic field. **b** Four bands (B1–B4) of magnetoresistance separated by three thresholds (i.e., 0.1%, 0.4%, and 0.7%), corresponding to different signaling modes in a). **c** Magnetoresistance response of sensor with different bent radii. Insets: photographs of bent sensor (top) and schematics of bent states (bottom). **d** Variation of magnetoresistance and electrical resistance of sensor stored in water as a function of time. Insets: magnetoresistance of sensor measured in air and water. **e** Sensor mounted on garment and zoom-in view of sensor in its original state, after cutting with micrometer-scaled crack, and after healing.

substrate or disconnection of composite trace (Fig. 5a), opening up possibilities for use as on-skin electronics, for example, training fingers for patients in medical therapy and rehabilitation. In this case, we attached a sensor to a finger and detected its relative position by reading out the resistance change as it moved with respect to a magnet mounted on the thumb of the same hand (Fig. 5b, Supplementary Fig. 26). Figure 5c records a magnetoresistance measurement as a function of time when the two fingers approached. By reading out the magnetoresistance signal, the real-time distance between two fingers can be monitored by doctors or patients themselves to decide on the progress of the training/rehabilitation (see Supplementary Fig. 27 for details).

## On-skin application for human–machine interface in augmented reality

On-skin magnetoresistive sensors are able to act as human–machine interfaces in augmented reality (AR), which could allow, for example, controlling the playback of a video with a pair of AR glasses (Supplementary Movie 7). Here, a physical interface consisting of a sensor and a magnet mounted on wooden thumb and forefinger, respectively, could control the information displayed on the lenses on the base of the magnetoresistance readout (Supplementary Fig. 28). Figure 5e records the magnetoresistance signal at different finger gestures

during such an interaction. In brief, by contacting two fingers for over 4 s (i.e., magnetoresistance exceeding a threshold), options popped up on the lenses of AR glasses (Fig. 5d$_{1-2}$); approaching and moving the fingers away (i.e., magnetoresistance increasing and decreasing) allowed scanning the options (Fig. 5d$_3$); holding (i.e., maintaining a magnetoresistance ratio) for over 2 s selected a particular option (Fig. 5d$_4$); and further approaching fingers (to cross the threshold) executed the selected program (Fig. 5d$_{5-6}$). The responsive speed of the magnetoresistive sensor is determined by the variation of the magnetic field, for example, altered on the timescale from milliseconds to seconds in Supplementary Fig. 29 and Supplementary Movie 8.

## Discussion

Different from conventional methods in which a percolation network is formed or healed passively by sediment or polymer mobility driving movement of fillers[11], here magnetic fillers aided by AMF can migrate deliberately, leading to the active formation or self-healing of percolation networks. In turn, the AMF-mediated printable magnetoresistive sensor achieved record performance and fulfilled six crucial criteria for practical self-healing. In light of the integration with textiles/skins and easy operation without the need of pre-alignment/pre-flattening, these sensors may find applications in fields as diverse as fitness training,

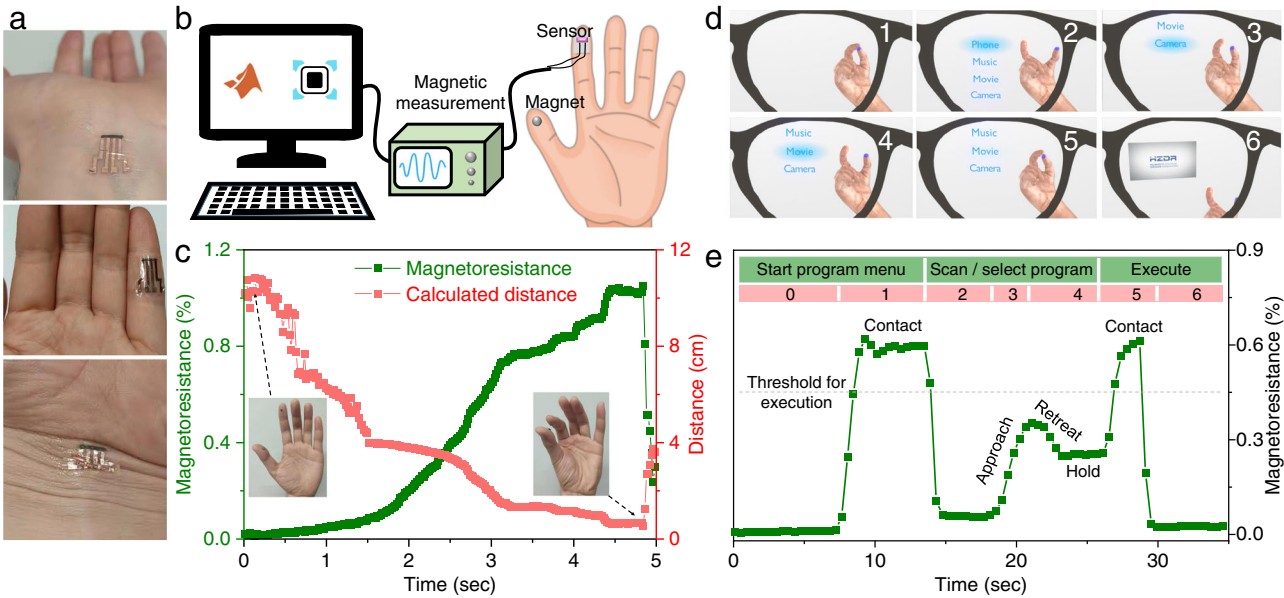

**Fig. 5 | On-skin printable magnetoresistive sensors for medical therapy and augmented reality. a** Photographs of sensors printed on 100-µm-thick plastic foil, conformally attached onto skin (from top to bottom) with planar, bent and crumpled surfaces. **b**, **c** Finger training application in medical therapy. **b** Schematic illustration of experimental configuration (from right to left): sensor and magnet compliantly adhering onto thumb and forefinger, magnetoresistance measurement, and analysis. **c** Magnetoresistance variation as two fingers were approaching and two-finger distance calculated from magnetoresistance readout. Inset: photographs of a thumb (magnet) and a forefinger (thumb) with different distances of about 10 cm and 1 cm. **d**, **e** Magnetoresistive sensor serving as human–machine interface in augmented reality (AR). **d** Screen shot for implementation of AR human–machine interface. Please see Supplementary Movie 7 for the concept realization. **e** Real-time magnetoresistance signal at different gestures to control AR glasses by sending various execution commands (e.g., start program menu, scan/ select program of interest, execute).

rehabilitation monitoring, human–machine interfaces, and wearable warning systems. Even if products are discarded, they can be effortlessly collected using permanent magnets and upcycled for circular economy.

Due to its processability and non-contact nature, the AMF-mediated method is compatible with and easily integrated into existing printing technologies. In fact, the results obtained in this work adapt to other printable magnetic devices, regardless of printing techniques, target substrates, polymeric binders (Supplementary Fig. 30), and magnetic fillers such as $Ni_{97}Co_3$ microparticles (Supplementary Fig. 31) and even surface-oxidized Fe microparticles, probably due to the fact that AMF could help to remove a thin natural oxide on the surface to allow for electrical percolation (Supplementary Fig. 32). This method will further benefit high-performance printable electronics with enhanced percolation by incorporating magnetic materials into fillers (e.g., forming mixtures or heterostructures), and give rise to reshapeable magnetic soft robotics by seamlessly splicing many components. It is foreseeable that the AMF-mediated self-healing also offers an attractive way for repairing implantable electronics by rapidly reforming percolation networks in vivo without the need for surgery to take out damaged electronics.

## Methods
### Fabrication and self-healing of printable magnetoresistive sensors
All chemicals for the composite synthesis are commercially available (Sigma Aldrich) including $Ni_{81}Fe_{19}$ powder, boric acid, hydroxyl-terminated polydimethylsiloxane (with viscosity of 18,000–22,000 cSt), n-hexane, isopropanol alcohol, and polydimethylsiloxane (184 silicone elastomer). The AMF for guiding formation and self-healing of percolation networks of $Ni_{81}Fe_{19}$ microparticles was generated by a lab-made solenoid with an induction coil and its details are introduced in Supplementary Fig. 4.

0.1 g polydimethylsiloxane (PDMS), consisting of base: curing agent = 10: 1, and supramolecular polyborosiloxane (PBS), formed by 0.9 g of hydroxyl-terminated poly(dimethylsiloxane) and 50 mg of boric acid, were used to form polymeric binder, as reported in ref. 29. PBS is a room-temperature self-healing polymer due to the dynamic boron/oxygen dative bonds at ambient conditions and the spontaneous entanglement of polymer chains[29,46,47], which is different from the polymers requiring external stimuli (e.g., illumination, heating, humidity) to initiate self-healing[48–50]. Notably, the PBS exhibits a "solid-liquid" viscoelastic behavior and cannot retain shape over long time scales[30]. By interpenetrating with the permanently cross-linked PDMS network, which serves as a structural scaffold to confine the flowability of PBS, the formed double-network polymer behaves as an elastic solid state and gains long-term structural stability and self-healing capability simultaneously (Supplementary Fig. 3)[29,32,47]. After the scaffold of the PDMS network is damaged, the PBS chains regain the intrinsic flowability. Subsequently, the PBS chains flow to the damaged interface and then the dynamic chemical bonds are reformed and the polymer chains are entangled, initiating the self-healing of polymers. With the assistance of AMF, the magnetic microparticles can be guided to reform the percolation networks, thus recovering the electrical conductivity. Considering the trade-off between the mechanical stability and the self-healing capability (namely, higher volume fraction of PDMS enhances the mechanical property, but deteriorates the self-healing capability; in contrast, higher volume fraction of self-healing PBS impairs the mechanical robustness), here we set the volume ratio between PBS and PDMS around 9:1 to balance the self-healing capability and mechanical robustness for the cured composites. 4 ml of mixture solvent (n-hexane: isopropanol alcohol = 1: 1) was added to reduce the viscosity of the binder solutions which prevents the agglomeration of $Ni_{81}Fe_{19}$ microparticles and leads to uniform dispersion of $Ni_{81}Fe_{19}$ microparticles (Supplementary Fig. 33a, b), thus favorable for the construction of electrically conductive

network. In the meantime, the low viscosity can effectively prevent the generation of air bubbles during agitating the binder solutions to disperse micropaticles (Supplementary Fig. 33c, d). The presence of air bubbles tends to block the electrical conductance in the printed trace. Moreover, the addition of solvent can precisely adjust the viscosity of the binder solution, consequently satisfying different viscosity requirements in various printing techniques. Magnetic stirring was applied to the mixture solution for about 30 min at room temperature to ensure the complete dissolving of all chemicals. By precise addition of $Ni_{81}Fe_{19}$ powders into binder solutions, composites with different concentration of magnetoresistive fillers were obtained for the following printing fabrication. Before printing, digital vortex mixer (VWR) was used to shake the composite solution for 30 s with a speed of 2500 rpm so that $Ni_{81}Fe_{19}$ microparticles inside could be uniformly dispersed in a whole volume of the solution. Three techniques including pipetting, spin coating, and screen printing were used for printing. A pipette controller (VWR) with volume of 100 μl was employed for pipetting. Spin coating was performed for 30 s at a speed of 500 rpm by WS-650-23 Spin Coater (Laurell). The surface coverage of the composite solution should be higher than 90%. Without particular notice, pipetting was used. Thanks to the intrinsic tackiness of the binder, the composite was capable of tightly adhering to different substrates. The as-synthesized composite was printed onto substrates of flat flexible cables, Si wafer, glass slides, plastic, paper, and ceramic. The sticky composite was placed into AMF for at least 10 s so that the magnetoresistive fillers can be magnetically guided to tighten contact between $Ni_{81}Fe_{19}$ microparticles and form electrically conductive matrix. To improve the mechanical stability during the cyclic bending/unbending test, a thick (e.g., of about 1–2 mm) conformal protective layer (made of the aforementioned PDMS and PBS with a ratio of 15:85) encapsulated the printed magnetoresistive composite trace to suppress the delamination of the composite from the substrate and/or the formation of cracking through the composite. Meanwhile, the active traces between measurement pads were set as 300 μm in length to decrease the structural deformation magnitude, given that the magnetoresistance is calibrated as the relative variation of the electrical resistance rather than the absolute value. Finally, the composite was heated at 120 °C for 12 h to cure the polymeric binders. Before the heating treatment, the solvent needs to be completely evaporated out of the composite (especially for the thick samples) by performing AMF for enough time or pumping vacuum, otherwise impairing the property of the cured composite and the self-healing performance (e.g., due to the generation of randomly dispersed bubbles). Depending on different fabrication parameters (e.g., solution loading, squeegee pressure, and spinning speed), the thickness of the cured composite varied from tens to hundreds of micrometers. Regarding the reference sample, the printed composite was placed into a uniform and static magnetic field of 500 mT for self-alignment. The composite trace was parallel with the magnetic field direction. The static magnetic field was generated by a lab electromagnet.

For self-healing, the solenoid-generated AMF was applied to damaged sensors at room temperature. Once exposed to AMF, $Ni_{81}Fe_{19}$ microparticles were magnetized along the field direction resulting in an attracting force between neighboring particles as discussed above. In seconds, such strong force led to physical touching between isolated fragments without manual reassembly, no matter for that with micrometer-scaled cracks and/or millimeter-scaled gaps. In the meantime, the alternating feature of AMF resulted in the mechanical oscillation of $Ni_{81}Fe_{19}$ microparticles, which could mill off a thin binder layer between $Ni_{81}Fe_{19}$ microparticles formed during self-healing and contributes to the reestablishment of electrical percolation networks. Due to the dynamic boron/oxygen dative bonds between neighboring chains in the PBS[29,30], the mechanical stability of polymeric binders could be healed at room temperature. Furthermore, the attraction and

oscillation of magnetic microparticles in AMF can assist mobility and intermixing of viscoelastic polymer binders, accelerating the mechanical healing of polymeric binders. However, the disconnected electrical path in the reference sample could not be healed at room temperature without the assistance of AMF, because heavy and dense network of $Ni_{81}Fe_{19}$ microparticles cannot be moved by the flow of viscoelastic polymer matrix. To strengthen the mobility of polymeric binds and $Ni_{81}Fe_{19}$ microparticles, thermal heating was performed over the damaged reference sample. The electrical paths were successfully reestablished for a minor fraction of samples after heating at 120 °C for tens of minutes (Supplementary Fig. 7).

### Dynamic mechanical analysis (DMA) measurements

We performed an amplitude sweep for the composites with different contents of polymer binders and an identical microparticles volume fraction of about 12.1% to define the linear viscoelastic region for further frequency sweep experiments. The test was performed at a constant angular frequency of 10 rad s$^{-1}$ in shear deformation range from 0.01% to 100% (Supplementary Fig. 3a). Based on the amplitude sweep results, 0.1% strain was chosen for the following frequency sweep experiment. The frequency sweep experiment, performed in linear viscoelastic region, does not introduce structural changes upon materials and allows for studying relaxation processes.

### Magnetoresistance measurement

Four-point configuration was used to measure the magnetoresistance of the printed magnetoresistive sensors. The gaps between measurement pads were about 0.3–1 mm. An electromagnet was used to generate uniform and tunable magnetic fields. The applied magnetic fields were swept between ± 12.5 mT. All measurements were carried out at room temperature. As of the operational durability test against repeated bending/unbending cycles, the printed sensor was manually deformed from the planar state to the bended state (Supplementary Fig. 23). After every five cycles of bending/unbending, the magnetoresistance was measured for the sensor at the planar state. A total number of 200 cycles was performed during the test. Before measurement 10 cycles of bending/unbending was pre-executed upon the sensor, followed by the AMF treatment to strengthen the $Ni_{81}Fe_{19}$ microparticle contact. The magnetic field direction was parallel with the electrical path of the magnetoresistive element of the sensor, unless otherwise noted.

### Static magnetic simulation

Stray fields of a homogeneously magnetized sample are uniform in the case of ellipsoidal or spherical shaped objects. The resulting magnetic field inside and outside an individual spherical $Ni_{81}Fe_{19}$ microparticle of radius, $r$, has the following form[51,52]:

$$\vec{B} = \begin{cases} -\frac{2}{3}\mu_0 \mathbf{M} + \mathbf{B}_{ext}, & \text{internal particle volume} \\ -\frac{\mu_0}{4\pi} \frac{3(\mathbf{M}\cdot\mathbf{r})\mathbf{r}-\mathbf{M}r^2}{r^5} + \mathbf{B}_{ext}, & \text{space outside particle} \end{cases} \quad (2)$$

where $\mu_0$ is the magnetic permeability of free space, $\mathbf{r}$ is the position vector, $\mathbf{M}$ is the magnetic moment of a particle, and $\mathbf{B}_{ext}$ is an external magnetic field. We calculate a spatial distribution of stray fields outside and inside two spherical microparticles with radius $r = 2$ μm (Fig. 3a$_1$). We consider two cases: (i) particles are placed at a distance of 850 nm from each other and (ii) particles stay in touch with each other. The resulting spatial field distributions for these cases are used for the calculation of magnetic forces acting on particles (Fig. 3a$_{2,3}$)[51,52]:

$$\mathbf{F} = (\mathbf{M}\cdot\nabla)\mathbf{B} \quad (3)$$

## Dynamic magnetic simulation

We numerically address a system of $N = 27$ spherical $Ni_{81}Fe_{19}$ particles of radius $r = 2$ μm and mass $m = 2.5 \times 10^{-4}$ μg each. We assume, that $i$th particle is always uniformly magnetized and creates a stray field

$$\mathbf{B}_i(\mathbf{r}) = \frac{\mu_0 M_S V}{4\pi}\left(3\frac{\mathbf{r}(\mu_i \cdot \mathbf{r})}{r^5} - \frac{\mu_i}{r^3}\right), i = \overline{1,N} \qquad (4)$$

where $M_S = 850$ kA/m is the saturation magnetization of $Ni_{81}Fe_{19}$, $V$ is the particle's volume and $\mu_i$ is the unit vector aligned with the magnetic moment of the particle. The motion of particles is governed by Newton's equations

$$m\ddot{\mathbf{r}}_i = \mathbf{F}_i(\mathbf{r}_i) - \mathbf{F}_s(\dot{\mathbf{r}}_i), i = \overline{1,N} \qquad (5)$$

where overdot indicates the derivative with respect to time $t$, $\mathbf{r}_i$ is the radius-vector of $i$th particle, $\mathbf{F}_i$ is the magnetic force

$$\mathbf{F}_i(\mathbf{r}) = M_S \sum_{j\neq i} \int \nabla\left(\mu_i \cdot \mathbf{B}_j(\mathbf{r})\right) dV, \qquad (6)$$

where the integration is performed over the volume of a particle taking into account a large stray field gradient from adjoining particles. The integration is performed via 32-point cubature expression for a ball[53]. The last term in (5) is Stokes's force

$$\mathbf{F}_s(\dot{\mathbf{r}}) = 6\pi R\eta\dot{\mathbf{r}} \qquad (7)$$

and $\eta = 1$ cSt being the dynamic viscosity of medium. Collisions between particles are simulated as partially inelastic with a reduction of the component of mechanical momentum along the collision direction by 90%.

To solve system (5), we utilize initial conditions, which include zero initial velocity for all particles, $\dot{\mathbf{r}}_i = \vec{0}$, and initial positions with the equal spacing of 7 μm randomly distorted within a range of 1.2 μm. The Euler scheme with the time step $\Delta t = 1$ ns is used for numerical integration. The initial directions of magnetic moments are slightly distorted from the direction of the external magnetic field.

Since the magnetization dynamics in $Ni_{81}Fe_{19}$ is characterized by the frequency of the ferromagnetic resonance of the order of 30 GHz, the temporal variation of $\mu_i$ can be considered as a "fast" variable in comparison with the mechanical degrees of freedom. To take this into account, the direction of $\mu_i$ at each time step is determined by the minimization of the following function:

$$\tilde{E}\left[\mu_i, i = \overline{1,N}\right] = \underbrace{\frac{\mu_0 M_S^2 V^2}{4\pi} \sum_{i\neq j}\left[\frac{\mu_i \cdot \mu_j}{r_{ij}^3} - \frac{3(\mu_i \cdot \mathbf{r}_{ij})(\mu_j \cdot \mathbf{r}_{ij})}{r_{ij}^5}\right]}_{E_{dip}}$$
$$\underbrace{-\mu_0 M_S V \sum_i \mu_i \cdot \mathbf{B}_{ext}(t)}_{E_{Zee}} \underbrace{+\Lambda N(1 - |\mu_i|^2)^2}_{E_{pen}} \qquad (8)$$

Here, $E_{dip}$ is the magnetic energy of interacting dipoles $\mu_i$, $E_{Zee}$ is the interaction of dipoles with the external magnetic field, and $E_{pen}$ is the penalty term preserving the unit length of $\mu_i$ with the penalty coefficient $\Lambda = 3000$. The energy minimization is done via NLopt package with the low-storage BFGS algorithm[54].

## Application of printable magnetoresistive sensors

(1) The algorithm of electrical signal processing in safety applications, demonstrated in Supplementary Movie 6 and Fig. 4a, b is summarized as follows: The magnetoresistance signal recorded from a printable and wearable magnetoresistive sensor by a USB-6211 data acquisition board (National Instruments, USA), was divided into four bands

(Fig. 4b). As the magnetoresistive sensor was far away from the permanent magnet (i.e., in Zone 1 of Fig. 4a), the magnetoresistance ratio was below the threshold of 0.1% and located in the B1 band. A low magnetic field around the sensor is due to environmental magnetic noise. In this magnetoresistance band, all indicating LEDs were off. As the permanent magnet approached (i.e., in Zone 2 of Fig. 4a), magnetic fields around the magnetoresistive sensor increase. The resultant magnetoresistance ratio crossed the threshold of 0.1% and raised into the B2 band below the threshold of 0.4%. Accordingly, the green LED was switched on. With further approaching into Zone 3 and Zone 4, the magnetoresistance ratio of the magnetoresistive sensor crossed the thresholds of 0.4% and 0.7% in sequence, and raised in the B3 and B4 bands, respectively. As a result, the yellow and red LEDs were tuned on respectively, signaling imminent health risk. The aforementioned logic required to define the magnetoresistance bands was programmed in LabVIEW (version 2019, National Instruments, USA).

(2) As of the application of finger training in medical therapy (Fig. 5b, c), a printable on-skin magnetoresistive sensor was mounted onto a forefinger, which interacted with a permanent magnet adhered to a thumb of the same hand. When approaching or retreating fingers, magnetic fields experienced by the magnetoresistive sensor changed accordingly, leading to the variation of magnetoresistance ratio. Based on the one-to-one correspondence between the finger distance and magnetoresistance ratio (Supplementary Fig. 27), the magnetoresistance signal was converted to a relative distance of fingers by numerical interpolation.

(3) For the application of human–machine interface (Fig. 5d, e), the manipulation of AR glasses was performed by executing commands based on the magnetoresistance signal of a printable on-skin magnetoresistive sensor. The magnetoresistance signal of the sensor mounted onto a wooden forefinger was tuned by the interaction with a permanent magnet adhered onto a wooden thumb (Supplementary Movie 7). The magnetoresistance ratio was divided into two regions that were separated by a threshold. As the magnetoresistance ratio was below the threshold, a list of program options was scanned by dynamically adjusting the magnetoresistance and the option was selected after remaining at a specific value for seconds. As the magnetoresistance ratio was above the threshold, the selected program was executed. To prove the functionality of the printable magnetoresistive sensor as being implanted beneath skin, a 3-mm-thick silicone rubber (ecoflex 00-50) was used to simulate the human skin, under which the magnetoresistive sensor was placed.

## Data availability

All of the data supporting the conclusions are available within the article and the Supplementary Information. Additional data are available from the corresponding authors upon reasonable request. Source data are provided with this paper.

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

## Acknowledgements

The authors are grateful to Dr. Ihor Veremchuk, Conrad Schubert, Nestor Miguel Valdez Garduño (all HZDR) for their help in the device preparation and characterization. The authors thank Dr. Nina Elkina (HZDR) for her support with numerical calculations, performed using the OpenStack and Hemera facilities at the High Performance Computing at HZDR. This work is financially supported in part by German Research foundation (DFG) grants MA 5144/13-1, MA 5144/28-1 and Helmholtz Association of German Research Centres in the frame of the Helmholtz Innovation Lab "FlexiSens".

## Author contributions

R.X. conceived the concept and performed experiments. O.V.P. and O.M.V. performed simulations. E.S.O.M. and P.M. carried out material and sensor characterization. G.S.C.B. and R.I. prepared the software and hardware for application demonstration. P.Mi. and L.I. performed dynamic mechanical analysis. R.X., Y.Z., G.S.C.B., and D.M. analyzed data. R.X., D.M., and G.S.C.B. wrote the manuscript and all authors edited it. D.M. and J.F. supervised the project.

## Funding

## Competing interests

The authors declare no competing interests.
