## [Peer Review File · Nature Communications]

Reviewers' Comments:

Reviewer #1:

Remarks to the Author:

The authors demonstrated a self-healable magnetic field sensor with the alternating magnetic fields (AMF)-driven percolation network that can be easily processable by various printing techniques. The developed sensor displays low noise and high-resolution performances in addition to the AMF-mediated self-healing performance with complete recovery, repeatable healing over a few of cycles, room-temperature operation, and humidity insensitivity. The suggested sensor consisting of microparticle-included polymer has the highest sensitivity and figure-of-merit for printable magnetoresistive sensors, which was used for applications such as wearable, biomedical, and human-machine interface. Although the sensing performance is superior to the other magnetoresistive sensors, the authors should address scientifically the characteristics of materials and self-healing performances. The reviewer recommends the publication of this manuscript in this journal after addressing the following comments.

1. For the construction of electrically conductive network, Ni₈₁Fe₁₉ microparticles would be homogeneously dispersed in the polymer matrix, which should be clarified with the scientific explanation. Supplementary Fig. 28 exhibited the microparticle dispersion in low viscosity solution without the consideration of surface interface and material property.
2. The self-healing mechanism is based on the dynamic boron/oxygen dative bond /double network, which has been already reported in Ref. 29, but driven by AMF. Considering the chemical structure of supramolecular poly-borosiloxane, the self-healing performance would be sensitive to the humidity. Since the authors explained humidity-insensitive self-healing performance, the reviewer suggests the self-healing property under water for the device in Supplementary Fig. 21.
3. AMF-mediated self-healing rate and performance is affected by the concentration of microparticles and the mechanical property of polymer composite, which should be provided.
4. In Figure 2a, the authors showed the magnetoresistance of sensors with different microparticle concentrations from 0.125 to 1.0 g/mL but not including 0.15 g/mL.
5. Many kinds of magnetoresistive sensors with self-healing ability have been reported. The authors are suggested to explain the novelty of this work in comparison with the reported self-healing polymers with the microparticles.

Reviewer #2:

Remarks to the Author:

This article by Makarov et al. reports on the printed magnetic field sensors fabricated by blending a polydimethylsiloxane scaffold, a supramolecular poly-borosiloxane self-healing polymer and Ni₈₁Fe₁₉ microparticles as fillers. The paste sensors show fast self-healing of both micrometer-scaled cracks and millimeter-scaled gaps under room temperature upon the alternating magnetic field treatment. The underlying mechanism is studied by combined computation and experiment approach. The alternating magnetic field is also found to assist the electrical percolation formation of fillers, which leads to the high sensitivity and low operational magnetic field. Finally, the authors present three examples of envisaged application to stress the performance advantage of the sensors. Overall this work is interesting and the manuscript is well organized. There are several technical concerns that need the authors' attention.

1. It is clear that the paste before curing is in a quasi-flowable state such that the microparticles can form percolation pathways under the applied magnetic field. And it is stated that after the paste printing, the material is heated to cure the polymer binder. But in the demo of the self-healing performance (Supplementary Movie 1 and 3), the material looks like remaining in a sticky state, which is confusing. If the cured polymer binder molecular chains are in a frozen solid state, how does the disconnected microparticle chains re-establish the conduction pathway after cracking? As stated in the manuscript, an insulating binder layer as thin as 1 nm is capable of blocking electrical path between microparticles, and the frozen molecular chains (not in sticky state) cannot be milled off by the oscillation of microparticles. But if the sensor material is always in a sticky tar-like state, is that viable for the proposed applications? It seems the manuscript does not involve such information. I suggest to clarify this point.
2. Following the last question, I am particularly confused about the term "cured paste". Mechanical

properties should be characterized to unveil if the cured material is indeed in the solid-like state or in the liquid-like state. This can be verified by comparing the storage modulus and loss modulus in a DMA test.

3. In the demonstration of safety monitoring, the distance between the magnet and the sensing device is unclear. What is the spatial range of Zone 1 – 4? And what is the spatial size shown in Supplementary Movie 6? Is this sensing distance dependent on the material composition and design?

4. Why the responsive speed of the sensor in the human-machine interface (Figure 5e) is slower than that in the safety monitoring (Figure 4b)? Is this responsive speed controllable?

5. Since this material is designed for wearable applications, mechanical fatigue property against repeated bending-unbending cycles is suggested to be added.

Point-to-point response to reviewers' comments & description of the change we have made to the manuscript to address these comments

Re: Manuscript ID NCOMMS-22-10151

We highly appreciate the detailed and constructive comments put forth by the reviewers, and have revised the manuscript accordingly, with all the reviewers' concerns being addressed. This letter details our point-by-point response to the comments. The revised text is written in blue font in our revised manuscript and supporting information.

Reviewer 1:

The authors demonstrated a self-healable magnetic field sensor with the alternating magnetic fields (AMF)-driven percolation network that can be easily processable by various printing techniques. The developed sensor displays low noise and high-resolution performances in addition to the AMF-mediated self-healing performance with complete recovery, repeatable healing over a few of cycles, room-temperature operation, and humidity insensitivity. The suggested sensor consisting of microparticle-included polymer has the highest sensitivity and figure-of-merit for printable magnetoresistive sensors, which was used for applications such as wearable, biomedical, and human-machine interface. Although the sensing performance is superior to the other magnetoresistive sensors, the authors should address scientifically the characteristics of materials and self-healing performances. The reviewer recommends the publication of this manuscript in this journal after addressing the following comments.

1. For the construction of electrically conductive network, Ni₈₁Fe₁₉ microparticles would be homogeneously dispersed in the polymer matrix, which should be clarified with the scientific explanation. Supplementary Fig. 28 exhibited the microparticle dispersion in low viscosity solution without the consideration of surface interface and material property.

To prove the homogeneous dispersion of microparticle fillers in the cured composite, we measured the electrical resistances. Supplementary Fig. 33e,f displays that the electrical resistances of two composite segments with 5 mm length are 112.3 and 108.2 Ω ,

respectively. The deviations from the mean resistance are about 2%, signaling the homogeneous dispersion of microparticles in the printed composite in consideration of the surface roughness variation of the plastic substrate. The scanning electron microscopy (SEM) image confirms no obvious agglomeration of microparticles in composite (Supplementary Fig. 33g). Statistical analysis demonstrates that the surface coverage of voids without microparticles are only 5.6%, and the voids are randomly distributed and isolated from each other, thus avoiding the blocking of electrical conductance along the printed composite trace. Note that, besides the prevention of microparticle agglomeration, the low viscosity solution with the addition of solvent is desirable for avoiding the generation of air bubbles (Supplementary Fig. 33d). The air bubbles generated in the high-viscosity polymeric binder as agitating to disperse microparticles in composites will block the formation of continuous electrical percolation networks (Supplementary Fig. 33c).

The explanation for homogeneous dispersion of microparticles in the polymer matrix is added on pages 38-39 of the revised Supplementary Information:

“As the polymeric binders of PDMS and PBS were dissolved in a solvent, the viscosity of the binder solutions can be tuned that is desirable for dispersion of $\text{Ni}_{81}\text{Fe}_{19}$ microparticles in the whole volume after shaking (Supplementary Fig. 33a). In contrast, as the binders were not diluted by the solvent, the composite exhibited very high viscosity. Consequently, $\text{Ni}_{81}\text{Fe}_{19}$ microparticles cannot be dispersed by shaking and always concentrated at the bottom of the composite (Supplementary Fig. 33b). It is worth noting that because of the high viscosity, air bubbles were easily generated in the binder without solvent during agitating that are not visually observed in the low-viscosity binder solutions diluted by solvent (Supplementary Fig. 33c,d). Therefore, the addition of solvent into the composite is crucial for forming the electrical percolation pathways in the following printing steps. Supplementary Fig. 33e, f compare the electrical resistances of two segments of a printed trace based on the diluted binders. Two resistance values for the same length of 5 mm are 112.3 and 108.2 Ω , respectively. Considering the roughness variation of the plastic substrate, the small resistance deviation of about 2% indicates the homogeneous dispersion of microparticles in the printed composite. The homogeneity of the microparticle dispersion can be further verified by SEM images of the printed composite in which no

obvious agglomeration of microparticles is observed (Supplementary Fig. 33g). Statistical analysis points out that voids without $\text{Ni}_{81}\text{Fe}_{19}$ microparticles only account for a 5.6% surface coverage. In particular, these voids are randomly distributed in the whole space and isolated from each other, thus avoiding the blocking of electrical conductance along the printed trace.”

Fig. S33 Magnetoresistive composite. Composites a) with and b) without solvents after shaking. Polymer binders c) with and d) without solvents after agitating. For a clear observation of the generated air bubbles, $\text{Ni}_{81}\text{Fe}_{19}$ microparticles are not mixed into the polymer binders. e,f) Electrical resistances for two segments of a printed composite trace. Insets: photography of the printed composite. g) SEM images of the printed composite after plasma etching. h) Voids without microparticles (denoted by red dots) derived from g). Statistically, the surface coverage of the voids is about 5.6%. Scale bars are 1 cm in a) – d) and 200 μm in g) – h), respectively.

2. *The self-healing mechanism is based on the dynamic boron/oxygen dative bond /double network, which has been already reported in Ref. 29, but driven by AMF. Considering the chemical structure of supramolecular poly-borosiloxane, the self-healing performance would be sensitive to the humidity. Since the authors explained humidity-insensitive self-healing performance, the reviewer suggests the self-healing property under water for the device in Supplementary Fig. 21.*

Thank you for the suggestion. In our case, the AMF treatment plays a crucial role for performing the self-healing in the water environment. Supplementary Fig. 25d and Supplementary Movie 3 shows that once exposing the damaged sensor to AMF, the attracting force generated from the magnetized fillers results in an *automatic* reconnection of two disconnected composites even in the water environment. It is noteworthy that the microscopic attraction force caused between the magnetized microscaled fillers are desirable for an intimate adhesion at the damaged interface without the production of small gaps. In the meantime, the water are squeezed out of the damaged interface of the composites, consequently avoiding the interference of water in the following reformation of chemical bonds. More importantly, the oscillation of Ni₈₁Fe₁₉ microparticles in AMF can strengthen the mobility of the polymer chains in the following self-healing. The improved mobility (with respect to the intrinsic mobility of polymers at room temperatures) not only accelerates the entanglement of polymer chains, but also provides more sites for chemical bond reformation that work in concert to complete the self-healing successfully. Supplementary Fig. 25c confirms that the damaged sensor regained its magnetoresistive response after AMF treatment in the presence of water.

The self-healing property of the magnetoresistive sensor in water is described on pages 6, 14 of the revised Manuscript and page 29 of the revised Supplementary Information:

“The AMF-mediated method also adapted to the reparation of splitting with millimeter-sized gaps even in humid environments (Fig. 1d). In Supplementary Movie 3, two magnetoresistive composites, soaked in water, were able to reconnect automatically within several milliseconds only with the assistance of the attracting force generated in the

magnetized $\text{Ni}_{81}\text{Fe}_{19}$ microparticles, beneficially omitting manual reassembly of disconnected composites^{9,11},” as stated on page 6 of the revised Manuscript.

“After damage, the magnetoresistive response can be healed even in water with the aid of AMF (Supplementary Fig. 25c,d), ascribed to the fast automatic reconnection, the intimate adhesion at the microscopic domain, and the enhanced mobility of the polymer chains.”, as stated on page 14 of the revised Manuscript.

Fig. S25 Magnetoresistive sensors in water. Sensors a) being placed and b) working in water. c) Magnetoresistance variation of a magnetoresistive sensor after carrying out cutting/healing in water for two times. d) Screenshots of Supplementary Movie 3, recording the AMF-mediated self-healing process of a damaged magnetoresistive composite in water (from left to right): d₁) water was poured into a beaker where the damaged composite was placed; d₂) the damaged magnetoresistive composite was completely soaked into water; d₃) two segments of the damaged composite was reconnected, driven by the AMF induced attracting force; d₄) self-healing was carried out through the dynamic reformation of chemical bonds and the entanglement of the polymer chains (driven by AMF-induced $\text{Ni}_{81}\text{Fe}_{19}$ microparticle oscillations); d₅) the damaged magnetoresistive composite was successfully healed in water. Scale bars: 1 cm.

3. AMF-mediated self-healing rate and performance is affected by the concentration of microparticles and the mechanical property of polymer composite, which should be provided.

Following the suggestion of the referee, we investigate the dependence of the AMF-mediated self-healing on the concentration of microparticles. To do so, we performed multiple cutting and healing at the same site upon sensors. We found that for the printing solutions with low microparticle concentrations (e.g., 0.125, 0.15, 0.25 g/ml, corresponding to 6.7%, 7.9%, 12.5% volume fractions of the microparticles in the cured composites without solvent), the printed sensors have low success rates (< 40%) for self-healing and the healed sensors demonstrate obvious reduction in magnetoresistance (Supplementary Fig. 9). As the microparticle concentration is increased to 0.5 g/ml (i.e., 22.3% volume fraction of the microparticles in composite), the self-healing rate is improved to 100% and the healed magnetoresistance show smaller variation. With further increasing the concentration to 1 g/ml (i.e., 36.5% volume fraction of the microparticles), the healed sample shows little magnetoresistance reduction. The difference of the self-healing rate and performance with the microparticle concentration may be ascribed to two factors: 1) the low concentration of microparticles results in a weak attracting force between the neighboring segments of the damaged sensors, probably leading to microscopic gaps at the damaged interface; 2) According to the percolation theory, the percolation threshold is about 16% volume fraction for spherical microparticles below which the percolation of the microparticles is considerably insufficient (DOI: 10.1038/NNANO.2012.192; 10.1201/9781315274386; 10.1146/annurev-matsci-070909-104529). Although the percolation pathways can be built below the threshold percolation of 16% in this work (e.g., 6.5%, 7.7%, 12.1% with the aid of the magnetically spatial alignment of $\text{Ni}_{81}\text{Fe}_{19}$ microparticles during AMF treatment), still the low volume fractions of microparticles adversely affects the reconnection of the damaged percolation paths. Apparently, increasing the volume fraction of the microparticles higher than the percolation threshold (e.g., to 22.3% and 36.5%) is desirable for enhancing the self-healing rate and performance.

The influence of the concentration of microparticles of polymer composite on the self-healing rate and performance is explained on page 8 of the revised Manuscript and page 11 of the revised Supplementary Information, and as follows.

“The concentration of $\text{Ni}_{81}\text{Fe}_{19}$ microparticles also affects the self-healing rate and the magnetoresistance performance (Supplementary Fig. 9). For instance, if the volume fraction of microparticles in the cured composite is lower than the ideal percolation threshold of about 16%^{12,40}, the healing of the damaged sensors was not reproducible. We anticipate that the reason for this poor self-healing performance is related to the presence of microscopic gaps at the damaged interface under weak attracting force and insufficient reconnection of percolation networks. With the addition of more microparticles to the composite (*e.g.*, to 22.3% and 36.5% volume fractions), the damaged sensors can be healed easily in AMF with little magnetoresistance reduction.”

Fig. S9 Self-healing of printed magnetoresistive sensors based on composites of different concentrations of $\text{Ni}_{81}\text{Fe}_{19}$ microparticles. The concentrations of $\text{Ni}_{81}\text{Fe}_{19}$ microparticles in the printing solutions (and the corresponding volume fractions in the cured composites) are a) 0.125 g/ml (6.7%), b) 0.15 g/ml (7.7%), c) 0.25 g/ml (12.1%), d) 0.5 g/ml (22.3%), e) 1g/ml (36.5%).

The mechanical property of a composite (and polymer binder) highly depends on the ratio of double polymeric networks, i.e., polydimethylsiloxane (PDMS) and poly-borosiloxane (PBS) in this work. Regarding the polymer mixing, one needs to balance a tradeoff between the mechanical robustness and the self-healing capability in the polymer system. On the one hand, Tang et al. reported that the polymer system with higher ratios of PDMS behaves more like an elastomer; as the ratio of PDMS is higher than 20%, the self-healing capability of the PDMS/PBS polymer will be impaired (DOI: 10.1039/C9TA09158K). Notably, incorporating fillers into the polymer will further decrease its self-healing capability (DOI: 10.1002/asia.202001157). Therefore, the 20% of PDMS should be the upper limit for remaining the high self-healing capability in composite. On the other hand, pure PBS (i.e., 0% ratio of PDMS) has no mechanical stability over long time scales (DOI: 10.1021/acsami.9b05230; 10.1002/adma.201501653). Based on these recognitions, it is concluded that the mixing ratios of around 10% PDMS used in our work should be an appropriate value to simultaneously maintain the self-healing capability and the mechanical robustness for the composite.

The influence of the mechanical property of polymer composite on the self-healing rate and performance (i.e., why we set the ratio of PBS/PDMS around 9:1) is explained on pages 19-20 of the revised Manuscript and as follows:

“Considering the trade-off between the mechanical stability and the self-healing capability (namely, higher volume fraction of PDMS enhances the mechanical property, but deteriorates the self-healing capability; in contrast, higher volume fraction of self-healing PBS impairs the mechanical robustness), here we set the volume ratio between PBS and PDMS around 9:1 to balance the self-healing capability and mechanical robustness for the cured composites.”

4. In Figure 2a, the authors showed the magnetoresistance of sensors with different microparticle concentrations from 0.125 to 1.0 g/mL but not including 0.15 g/mL.

The magnetoresistance data of sensors with 0.15 g/ml microparticles has been added to Fig. 2a, shown as follows and on the page 10 of the revised Manuscript.

Figure 2a. Magnetoresistance of sensors made by composites of different $\text{Ni}_{81}\text{Fe}_{19}$ microparticle concentrations.

5. Many kinds of magnetoresistive sensors with self-healing ability have been reported. The authors are suggested to explain the novelty of this work in comparison with the reported self-healing polymers with the microparticles.

We appreciate this remark of the referee. Although the electronic sensors with self-healing ability have been widely reported, few works are focused on magnetoresistive (or magnetic field) sensing. We guess the reviewer had suggested for the comparison to the reported self-healing polymers with the microparticles in other electronic sensors. In the following, we will explain the novelty of this work from this perspective.

For an electronically active composite system, consisting of polymeric binders and functional fillers, the ideal self-healing after damage includes the recovery of the morphology/mechanical strength and the electrical functionality. The former is determined

by the property of polymers, that is, the dynamic reformation of chemical bonds and/or the entanglement of polymer chains at the damaged sites; the latter depends on the reestablishment of electrical percolation pathways of functional fillers.

Regarding the polymer healing, regenerating the bonds in an easy way and enhancing polymer chain diffusion are beneficial for a successful self-healing (DOI: 10.1002/adma.201003036, 10.1039/C3CS60109A, 10.1002/admi.201800384, 10.1002/adma.201604973). For some self-healing polymers, external stimuli are needed for the regeneration of broken chemical bonds, *e.g.*, illumination (DOI: 10.1038/nature09963), heating (DOI: 10.1039/C2PY20957H), and humidity (DOI: 10.1021/acs.macromol.5b00210, 10.1002/adma.201201306). The polymers applied in this work (*i.e.*, polyborosiloxane) is a room-temperature self-healing polymer, namely, the association or dissociation of bonds between adjacent polymeric chains can occur spontaneously at ambient conditions and the polymer chains can be entangled without the assistance of external stimuli (DOI: 10.1021/acsami.6b06137). More importantly, the proposed alternating magnetic field (AMF) treatment could cause the oscillation of functional fillers (here, Ni₈₁Fe₁₉ micropraticles) in the composite. Accordingly, these controllable oscillations of fillers (in intensity, frequency, direction, and time by tuning AMF generator, we refer to Supplementary Fig. S4) are able to strengthen the movement and thus the entanglement of polymer chains because of the sticky adhesion at the filler/polymer interfaces. In the meantime, this continuous mixing of polymers also provide more opportunities for the broken chemical bonds to contact with each other. The above two effects work complementarily to accelerate the self-healing process of polymers.

Although many self-healing polymers have been developed, only a limited number of self-healing composite systems have been used in electronics. One of the main hinders is a lack of an effective way to establish and reestablish the electrical properties (*e.g.*, electrical conductivity and magnetoresistance). In general, the method for reestablishing electrical percolation networks in the composite can be categorized into three aspects: **1)** releasing healing agents (conductive materials or assistant materials) from microcapsules at the damaged sites (DOI: 10.1002/adma.201102888, 10.1002/adma.201200196). However, only a limited number of healing times can be carried out, and normally healing cannot be

repeated at one site, due to the dissipation of healing agents. In our case, the self-healing process does not require the assistance of healing agents and thus can be repeatedly performed through using AMF to rearrange magnetic fillers and repair the broken electrical percolation networks. **2)** Internal stimuli originating from the polymers, that is, the mobility of polymeric binders driving the movement of fillers to the damaged sites (DOI: 10.1038/s41565-018-0244-6, 10.1021/acsami.6b06137). However, the driving force arising from the polymer mobility may not be strong enough to move the dense and heavy fillers, leading to failure in 100% restoration of electrical property. Although heating the polymers can enhance the movement and provide stronger driving force, but the high temperature may harm surrounding materials and devices. In our case, the driving force over the fillers was generated by the AMF and can be easily controlled in magnitude, direction, frequency and time. The AMF-induced force is much stronger than the force caused from the polymer mobility even at high temperature. In particular, as compared with the spatially random force caused by the mobility of polymers, the AMF-induced directional force, e.g., vertical to the damaged interface, is more effective for guiding the fillers to the damaged sites. **3)** External stimuli exerted onto the fillers, e.g., dielectrophoresis-assisted alignment of fillers (DOI: 10.1002/adma.202001642), magnetic-field induced attracting force by incorporating permanent magnets into composites (DOI: 10.1126/sciadv.1601465). However, the electrical properties healed by dielectrophoresis-assisted alignment of fillers cannot be 100% restored, e.g., the resistance of the healed samples were always higher (with an average increase of 35.6%) than that of the pristine. Although the method by incorporating permanent magnets into composites could realize automatic adhesion after damage, the strong magnetic field generated around the permanent magnets might interfere with the function of nearby electronics and is harmful to the health of humans for long-time exposure. Even worse is the magnetic hysteresis phenomenon that prevents the utilization of the permanent magnets as magnetoresistance fillers. In contrast, the AMF-induced self-healing not only realizes the automatic reconnection of damaged composites aided by the magnetized $\text{Ni}_{81}\text{Fe}_{19}$ microparticles but also remains zero net magnetic field around the composite after removing AMF (and thus has no negative side effects). Due to the guidance of AMF over functional fillers, the AMF-assisted self-healing can 100% recover the electrical conductivity.

In comparison with the as-reported self-healing polymers with microparticles, the novelties of this work, that are summarized in the preceding paragraphs but not mentioned in the previous version of manuscript, are explained on pages 5, 6, 19 of the revised Manuscript and as follows:

‘Besides reforming conductive pathways like that in conventional methods (*e.g.*, by taking advantage of the mobility of polymers which drives the movement of fillers)^{9,11}, the AMF-mediated self-healing could 100% recover the original electrical property in few seconds due to the controllable magnetic force (*e.g.*, strength and direction of the applied magnetic field, its frequency, and the actuation time) applied to Ni₈₁Fe₁₉ microparticles. In particular, the AMF-generated force is independent of the surrounding conditions (*e.g.*, temperature and humidity), thus broadening the applicability of the AMF-induced self-healing.’ as described on page 5 of the revised Manuscript.

“The AMF-mediated method also adapted to the reparation of splitting with millimeter-sized gaps even in humid environments (Fig. 1d). In Supplementary Movie 3, two magnetoresistive composites, soaked in water, were able to reconnect automatically within several milliseconds only with the assistance of the attracting force generated in the magnetized Ni₈₁Fe₁₉ microparticles, beneficially omitting manual reassembly of disconnected composites^{9,11}. In stark contrast to the devices that incorporate permanent magnets as fillers for automatic reconnection in self-healing, the AMF-mediated self-healing sensor has no magnetic remanence and thus will not pose risk to human health and/or interfere with the functionality of nearby electronics^{38,39}. In particular, as compared with the manual reconnection at the macroscale (during which microscopic gaps might be retained at the damaged interface), the strong attracting force triggered by the magnetized microparticles could result in an intimate adhesion at the microscopic domain, which is beneficial for 100% healing of the electrical performance.” as described on page 6 of the revised Manuscript.

“PBS is a room-temperature self-healing polymer due to the dynamic boron/oxygen dative bonds at ambient conditions and the spontaneous entanglement of polymer chains^{29,53,54},

which is different from the polymers requiring external stimuli (*e.g.*, illumination, heating, humidity) to initiate self-healing⁵⁵⁻⁵⁷.” as described on page 19 of the revised Manuscript.

Reviewer 2:

This article by Makarov et al. reports on the printed magnetic field sensors fabricated by blending a polydimethylsiloxane scaffold, a supramolecular poly-borosiloxane self-healing polymer and Ni₈₁Fe₁₉ microparticles as fillers. The composite sensors show fast self-healing of both micrometer-scaled cracks and millimeter-scaled gaps under room temperature upon the alternating magnetic field treatment. The underlying mechanism is studied by combined computation and experiment approach. The alternating magnetic field is also found to assist the electrical percolation formation of fillers, which leads to the high sensitivity and low operational magnetic field. Finally, the authors present three examples of envisaged application to stress the performance advantage of the sensors. Overall this work is interesting and the manuscript is well organized. There are several technical concerns that need the authors' attention.

1) It is clear that the composite before curing is in a quasi-flowable state such that the microparticles can form percolation pathways under the applied magnetic field. And it is stated that after the composite printing, the material is heated to cure the polymer binder. But in the demo of the self-healing performance (Supplementary Movie 1 and 3), the material looks like remaining in a sticky state, which is confusing. If the cured polymer binder molecular chains are in a frozen solid state, how does the disconnected microparticle chains re-establish the conduction pathway after cracking? As stated in the manuscript, an insulating binder layer as thin as 1 nm is capable of blocking electrical path between microparticles, and the frozen molecular chains (not in sticky state) cannot be milled off by the oscillation of microparticles. But if the sensor material is always in a sticky tar-like state, is that viable for the proposed applications? It seems the manuscript does not involve such information. I suggest to clarify this point.

We thank the referee for his/her suggestion. For a clear explanation, we firstly give a brief introduction into the self-healing mechanism of the applied self-healing polymer system and then answer this question.

The self-healing mechanism of the applied self-healing polymer system:

The polymeric binder of the composite used in this work belongs to a class of double-network structured polymers. Essentially, such polymers, consisting of two interpenetrating networks, inherit different functionalities to from a brand-new material with the combination of the physical and chemical advantages of two components (DOI: 10.1002/adfm.202110244; 10.1002/adma.202003155; 10.1002/macp.201600038), for example, gaining self-healing capability and mechanical stability into the polymer binder in this work. Among diverse double-network polymers, the polyborosiloxane/polydimethylsiloxane (PBS/PDMS) system is attracting attentions (DOI: 10.1039/C9TA09158K; 10.1021/acsami.1c00282; 10.1002/mame.202000621; 10.1021/acsami.9b05230; 10.1021/acsami.6b06137, etc). The component PBS features an intrinsic self-healing property due to both the dynamic chemical bonds (including boron/oxygen dative bonds and hydrogen bonds) and the entanglement of adjacent polymer chains (DOI: 10.1021/acs.iecr.6b03823; 10.1039/c8tc01092g; 10.1021/acsami.6b06137). However, the PBS with the “solid-liquid” viscoelastic behavior is structurally unstable over long time scales, and will flow as a high-viscosity liquid without the possibility to recover its original morphology (DOI: 10.1021/acsami.9b05230; 10.1002/adma.201501653). By interpenetrating the permanently cross-linked PDMS network within the dynamically cross-linked PBS network, the obtained material gains two advantages of long-term structural stability and self-healing capability simultaneously (DOI: 10.1039/C9TA09158K; 10.1021/acsami.1c00282; 10.1002/mame.202000621). In other words, the presence of the permanent PDMS network, serving as a scaffold, confines the flowability of PBS to maintain the structural stability of the double-network matrix and imparts mechanical toughness. Accordingly, the double-network polymers behave as an elastic solid state; once being damaged, the unconfined PBS at the local site of damages flows out of the PDMS scaffold. Then, dynamic chemical bonds of PBS are reformed and polymer chains are entangled, consequently initiating self-healing of polymers. In the meantime, the mobility of PBS may afford driving force for the functional fillers to move toward the damaged interface and rebuild the electrical percolation pathways (DOI: 10.1039/C9TA09158K; 10.1021/acsami.1c00282; 10.1002/mame.202000621; 10.1021/acsami.9b05230; 10.1021/acsami.6b06137).

The answer to this question is explained in the following:

Based on the above description, it can be concluded that the cured composite behaves as follows: before damage, the composite is in the elastic solid state with long-term mechanical robustness (caused by the formation of two interpenetrating matrix of permanently cross-linked PDMS network and the dynamically cross-linked PBS network), rather than a sticky tar-like state or a frozen solid state. After damaging the PDMS scaffold, viscoelastic PBS leaks out at the damaged interface as well as the microparticles that initiate the following self-healing. Because of the elastic solid behavior, the cured composite can be used as functional components of the electronic devices. To date, the polymer systems with the similar property have been widely used for the fabrication of soft electronics as well as the self-healing of the electrical property after damage (DOI: 10.1038/s41928-019-0235-0; 10.1021/acsami.7b19511; 10.1016/j.progpolymsci.2013.08.001; 10.1002/adma.201604973; 10.1002/adma.202004190; 10.1002/adma.201904765). However, limited by the difficulty in the electrical percolation network formation during fabrication and the percolation network reformation after damage (as stated in the manuscript), most of the devices are based on the fillers with nanoscaled structures (DOI: 10.1126/science.aag2879; 10.1038/nnano.2012.192; 10.1038/s41565-018-0244-6; 10.1021/acsami.1c00282; 10.1002/adma.201501653; 10.1021/acsami.9b05230; 10.1039/C9TA09158K; 10.1145/3332165.3347901; 10.1088/1361-6528/abe6c7; 10.1109/ECTC32862.2020.00350; 10.1016/j.cej.2021.128734). For example, the driving force caused from the mobility of polymer chains is usually not strong enough to move heavy fillers for self-healing, leading to suppression in electrical performance of the healed devices to different extents. As stated in the answer to Question 5 of Reviewer 1, the AMF force applied upon $\text{Ni}_{81}\text{Fe}_{19}$ microparticles can be precisely tuned in intensity, direction, frequency, and time, thus beneficial for peroration network (re)formation with the following advantages: 1) due to the force intensity controllability, the heavy and dense $\text{Ni}_{81}\text{Fe}_{19}$ microparticles can be easily driven to move and oscillate inside composites; 2) due to the force direction controllability, the $\text{Ni}_{81}\text{Fe}_{19}$ microparticles can be navigated to the damaged interface directly, introducing more fillers for percolation pathway reformation and accelerating the self-healing process; 3) due to the force frequency controllability, the microparticle oscillation can be tuned to mill off a thin isolating layer between microparticles; 4) the

self-healing time can be tuned until the ideal self-healing performance is obtained. It is noteworthy that the elastic solid-state property of the cured composite benefits for the intimate adhesion at microscopic domains of the damaged interface (DOI: doi.org/10.1002/adma.200901940; 10.1039/C9NR09438E). As compared with the macroscale reconnection generated by manual treatment (during which microscopic gaps might be retained at the damaged interface), the microscopic intimate adhesion between damaged composite, caused from the microscale magnetic interaction between Ni₈₁Fe₁₉ microparticles, plays an important role in 100% healing the electrical performance.

The explanation about the composite state and the corresponding percolation (re)formation is stated on page 19 of the revised manuscript:

“By interpenetrating with the permanently cross-linked PDMS network, which serves as a structural scaffold to confine the flowability of PBS, the formed double-network polymer behaves as an elastic solid state and gains long-term structural stability and self-healing capability simultaneously (Supplementary Fig. 3)^{29,32,54}. After the scaffold of the PDMS network is damaged, the PBS chains regain the intrinsic flowability. Subsequently, the PBS chains flow to the damaged interface and then the dynamic chemical bonds are reformed and the polymer chains are entangled, initiating the self-healing of polymers. With the assistance of AMF, the magnetic microparticles can be guided to reform the percolation networks, thus recovering the electrical conductivity.”

2) Following the last question, I am particularly confused about the term “cured composite”. Mechanical properties should be characterized to unveil if the cured material is indeed in the solid-like state or in the liquid-like state. This can be verified by comparing the storage modulus and loss modulus in a DMA test.

According to the suggestion of the referee, the DMA test of the cured composite was performed and the mechanical properties are explained on page 4 of the revised Manuscript and on pages 4, 5 of the revised Supplementary Information, and as follows:

“The cured composites exhibit higher storage moduli than loss moduli over the whole range of frequency (Supplementary Fig. 3), revealing an elastic behavior and mechanical stability (*i.e.*, no flowing) over long timescales which is in agreement with previous reports^{29,32,33}.” as stated on page 4 of the revised Manuscript.

“The results confirm the elastic behavior of all samples, given that the storage moduli are higher than loss moduli over the entire frequency range. The elasticity for the composite made of pure PDMS is due to covalently crosslinked polymer networks. The addition of high amount of PBS introduces the relaxation processes of the composite around 1 rad/s. This behavior is caused by the relaxation of supramolecular network of PBS. The higher amount of PBS, the more pronounced the effect of PBS relaxation. Pure PBS behaves as viscous liquid at long time scale and flows. Fortunately, even 10% of crosslinked PDMS can provide mechanical stability². Note that, the microparticle volume fractions of the measured composites are only about 12%. With the addition of more microparticles, the mechanical stability can be further enhanced due to the confinement of microparticles matrix³.” as stated on pages 4, 5 of the Supplementary Information.

Fig. S3 Dynamic mechanical analysis (DMA) of cured composites. a) Amplitude sweep characterization (at a frequency of 10 rad/s) for the composites with different polymer contents (from left to right): PDMS : PBS = 10 : 90, PDMS : PBS = 20 : 80, pure PDMS. b) Viscoelastic behavior characterization at small deformations.

3) *In the demonstration of safety monitoring, the distance between the magnet and the sensing device is unclear. What is the spatial range of Zone 1 – 4? And what is the spatial size shown in Supplementary Movie 6? Is this sensing distance dependent on the material composition and design?*

We will answer this question point-by-point:

“Is this sensing distance dependent on the material composition and design?”

The sensing distance of a magnetoresistive sensor is dependent on two aspects: 1) the sensing capability of the sensor (i.e., magnetoresistance as a function of magnetic field) and 2) magnetic field distribution generated around the permanent magnet or electromagnet, etc.

Regarding the magnetoresistance of a printed sensor, it is determined by both the magnetoresistive fillers in the composite (e.g., Ni, Fe, Co, alloy) and the electrical percolation network of these fillers (i.e., high quality of percolation networks results in high magnetoresistance response). In this work, we employed Ni₈₁Fe₁₉ microparticles as the magnetoresistive fillers, because they are commercially available and have been widely investigated in the magnetic performance. We then optimized the electrical percolation network of the magnetoresistive fillers by performing the AMF treatment.

For a specific magnetoresistive sensor (e.g., the AMF-mediated sensors in our work), the sensing distance relies on the magnetic field intensity and distribution in the proximity of the magnet. For a strong magnet, high magnetic fields will be produced and thus the sensing distance will be long; while for a weak magnet with low magnetic fields, the sensing distance of the sensor will be shorter (please refer to Supplementary Fig. 19).

‘In the demonstration of safety monitoring, the distance between the magnet and the sensing device is unclear. What is the spatial range of Zone 1 – 4?’

In the demonstration of safety monitoring, the sensing capability of the applied sensor (i.e., the magnetoresistance curve with magnetic field) is specified. Accordingly, the magnetoresistance thresholds (i.e., 0.1%, 0.4%, and 0.7%) correspond to 0.06, 0.2, and 1.3

mT, respectively (Fig. 4d). Therefore, the sensing distance is determined by the magnetic property of the magnet (i.e., the magnetic field intensity and distribution). Here, we take two magnets as an example (Supplementary Fig. 19).

The magnetic fields as a function of the distance apart from the magnet are plotted in Supplementary Fig. 19b-e. Obviously, two magnets have different magnetic field distributions. For a large cube-shaped magnet in Supplementary Fig. 19b,c, the sensing distances (namely, the boundaries of Zone 1→2, 2→3, and 3→4 with magnetic fields of 0.061, 0.208, 1.32 mT) are about 63, 50, 27 cm, respectively. For a ring-shaped magnet in Supplementary Fig. 19d,e, the sensing distances are about 46, 31, 15 cm, corresponding to the magnetic fields of 0.06, 0.20, 1.28 mT), respectively.

And what is the spatial size shown in Supplementary Movie 6?

The spatial scale bar has been added to Supplementary Movie 6. To clearly show all components (LED indicator, magnetoresistive sensor, magnet) in the video, here we used a small magnet (dimension: 4 mm × 4 mm × 4 mm, magnetic field at surface: 33.2 mT in Supplementary Fig. 19a) to conceptually exhibit how the magnetoresistive sensor works. The same magnet is also applied in Supplementary Video 8 to show the responsive speed of the sensor.

The spatial ranges of Zone 1 – 4 are described on page 13 of the revised Manuscript:

“Depending on the magnetic stimulus (*e.g.*, magnetic field intensity and its spatial distribution), the sensing zones are altered for different magnets. For example, the boundaries between Zone 1, Zone 2, Zone 3 and Zone 4 are 63, 50, and 27 cm (Supplementary Fig. 19b,c) as well as 46, 31, and 15 (Supplementary Fig. 19d,e) away from the corresponding magnets.”

Fig. S19 Magnetic fields around permanent magnets. a) Commercial permanent magnets used for the characterization of the printed magnetoresistive sensors in the safety monitoring. From left to right, a large cube-shaped magnet is of about $10\text{ cm} \times 10\text{ cm} \times 2\text{ cm}$ in dimension and 300 mT in magnetic field at its surface; a ring-shaped magnet has 10

cm outer diameter, 8.5 cm inner diameter, 7 mm thickness, and 296 mT magnetic field; a small cube-shaped magnet is of 0.4 cm × 0.4 cm × 0.4 cm in dimension and 33.2 mT in magnetic field. Magnetic field distribution in the proximity of b,c) the large cube-shaped magnet and d,e) the ring-shaped magnet. Based on the magnetoresistance curve of the sensor in Fig. 4d, the magnetoresistance thresholds of 0.1% (corresponding to the boundary of Zone 1 and Zone 2), 0.4% (corresponding to the boundary of Zone 2 and Zone 3), and 0.7% (corresponding to the boundary of Zone 3 and Zone 4) are about 0.06, 0.2, and 1.3 mT. Accordingly, the spatial configuration of Zone 1-4 are marked in c,e). The magnetic field at the corresponding boundaries, measured by a magnetic field tester, is also exhibited in b,d) for a clear understanding of the distance dependence of the magnetic field apart from the permanent magnet. Scale bars are 2 cm in a) and 5 cm in b,d).

4) Why the responsive speed of the sensor in the human-machine interface (Figure 5e) is slower than that in the safety monitoring (Figure 4b)? Is this responsive speed controllable?

The magnetoresistance of the magnetoresistive sensor is determined by the magnetic field experienced. In other words, the responsive speed of the sensor only depends on the variation speed of the magnetic field around the sensor. For instance, the responsive speed can be varied from 2 seconds (i.e., the 5th pulse in Supplementary Fig. 29a) to 100 ms (i.e., the 3rd pulse in Supplementary Fig. 29a), due to different velocities that the magnet approaches and leaves away from the sensor (please refer to Supplementary Video 8). Considering the different velocities that the permanent magnets interacted with the sensors in the human-machine interface (Fig. 5e) and in the safety monitoring (Fig. 4b), the responsive speed of the sensor varied accordingly.

The responsive speed of the magnetoresistive sensor is explained on page 17 of the revised Manuscript and as follows:

“The responsive speed of the magnetoresistive sensor is determined by the variation of the magnetic field, for example, altered on the timescale from milliseconds to seconds in Supplementary Fig. 29 and Supplementary Movie 8.”

Fig. S29 Demonstration for the response speed of the printed magnetoresistive sensor. a) Experimental setup (including a monitor, a permanent magnet, and the magnetoresistive sensor) and the magnetoresistance responsive curve with the magnet approaching the sensor for five times at different speeds. The third pulse of the magnetoresistance curve in a) are plotted in b) for clear observation. Scale bar: 3 cm.

5) *Since this material is designed for wearable applications, mechanical fatigue property against repeated bending-unbending cycles is suggested to be added.*

Following the suggestion of the referee, we measured the magnetoresistance variation of the magnetoresistive sensor against repeated bending/unbending cycles to verify the durability for wearable applications. To enhance the mechanical robustness, a conformal protective layer (e.g. of about 1 mm and made of the same polymeric binder, please refer to Methods for details on page 21 of the revised Manuscript) was coated on the sensor to suppress the delamination and/or the crack formation of conductive layers during continuous deformation. It is found that the printed magnetoresistive sensor has reliable mechanical durability against cyclic deformation (Supplementary Fig. 23). After 200 bending/unbending cycles, the magnetoresistance ratios still can be maintained over 0.9%.

The operational durability of the printed magnetoresistive sensor against repeated bending-unbending cycles is described on page 14 of the revised Manuscript and as follows:

“Besides, the AMF-treated sensor has reliable operational stability (Supplementary Fig. 23). After hundreds of bending/unbending cycles, the high magnetic sensing capability was

maintained. Notably, the magnetoresistance suppression induced by the cyclic deformation can be easily healed by the AMF treatment.”

Fig. S23 Operational stability test for the printed magnetoresistive sensor against repeated bending and unbending. a) Magnetoresistance of the printed sensor after repeated bending/unbending cycles. The labels indicate the number of bending/unbending cycles. b) Magnetoresistance of the printed sensor at 12.5 mT as a function of bending/unbending cycles. The magnetoresistances were derived from a). Inset: photographs for the printed sensor in the planar state (right) and the bended state (right). Scale bars: 1 cm.

Reviewers' Comments:

Reviewer #1:

Remarks to the Author:

Authors fully addressed all the comments raised by the reviewers.

Reviewer #2:

Remarks to the Author:

In this revision, the authors have addressed all the comments from the reviewer. The revised manuscript is now suitable for publication in Nat. Commun.